# Shear Strengthening by GFRPU on an RC Column with Nonseismic Reinforcement Details

**DOI:** 10.3390/polym14173596

**Published:** 2022-08-31

**Authors:** Eun-Taik Lee, Jae-Hyoung An, Jun-Hyeok Song, Yu-Sik Hong, Hee-Chang Eun

**Affiliations:** 1Department of Architectural Engineering, Chung-Ang University, Seoul 06974, Korea; 2Department of Architectural Engineering, Kangwon National University, Chuncheon 24341, Korea

**Keywords:** glass fiber reinforced polyurea, polymer, retrofit, cyclic load, energy dissipation, shear strengthening, ductility

## Abstract

Structures constructed before seismic design standards in Korea were enacted are being seismically and structurally strengthened for structural performance. The jacket method has been used to enhance the shear load-carrying capacity and ductility of reinforced concrete columns. This study investigated the effect of glass fiber-reinforced polyurea (GFRPU) on the enhancement of shear strength and ductility. It was shown that the GFRPU spraying technique played an important role in enhancing the column because of the material characteristics of the GFRPU, that is, high-tensile strength and elongation. The reinforcement effect of GFRPU on the shear-span ratio and axial-force ratio of six reinforced concrete (RC) column specimens was evaluated. The experimental results demonstrated that the lateral-reinforcement effect and energy-dissipation capacity were improved. The H-series and S-series specimens reinforced by the GFRPU with an axial-load ratio of 0.1 exhibited higher shear strength, of about 30% and 18%, than the specimens without reinforcement. The shear-load-carrying capacity of the S-series specimen with the axial-load-force ratio of 0.2 increased by 4%. The design method representing the lateral-resistance ability of the GFRPU was represented by an empirical formula reflecting the experimental results.

## 1. Introduction

Reinforced concrete (RC) structures deteriorate due to environmental factors, the period of use, natural disasters, excessive loads, or repetitive external factors. In Korea, earthquake-resistance design standards were enacted in 1987. Therefore, structural and seismic performance reinforcements need to be applied to structures constructed before this period. Nonseismic RC columns with nonseismic reinforcement details are vulnerable to shear failure and loss of axial strength. Several shear reinforcement methods have been proposed to secure load-carrying capacity and ductility to improve structural or seismic performance.

Three types of strengthening methods including RC jacketing, steel jacketing, and fiber-reinforced polymer (FRP) confining or jacketing have been utilized. FRP is a composite material made of a polymer matrix reinforced with fibers such as glass, carbon, or aramid. The FRP strengthening approach is the most widely-used method. The method improves the ultimate load-carrying capacity, shear capacity, and ductility of reinforced concrete columns.

FRP improves the shear capacity, ductility, and energy-dissipation capacity of RC columns. Ye et al. [1] proposed a shear coefficient to evaluate the shear contribution of a carbon fiber-reinforced polymer (CFRP) sheet by using the following parameters: CFRP sheet amount, shear-span ratio, and axial-load ratio. Kurniawan et al. [2] observed an enhancement of the peak load and energy dissipation of RC columns strengthened with CFRP. Dionysios et al. [3] investigated a retrofitting method to enhance flexural resistance that combined epoxy-bonded near-surface mounted reinforcing materials with textile-reinforced mortars. Bedirhanoglu and Triantafillou [4] investigated the seismic behavior of FRP-retrofitted low-strength RC short column specimens. Troung et al. [5] evaluated various retrofit methods for RC columns with nonseismic reinforcement details and compared two retrofit strategies, namely a partial retrofit in the plastic hinge zone and a full retrofit of the entire column. Yoddumrong et al. [6] studied the strengthening performance of low-cost glass fiber-reinforced polymers (GFRP). Shin and Jeon [7] proposed a retrofit technique to mitigate the seismic vulnerabilities that used additional confinement and increased the stiffness of existing columns. Amin and Ahmad [8] assessed the structural performance of a building structure strengthened with multilayer CFRP jacketing. Shaoqing and Kono [9] investigated the shear contribution of the FRP sheets to prevent the brittle fracture of the column. Li et al. [10] developed a hybrid fiber reinforced polymer (HFRP) to replace the steel cables and investigated the failure modes of anchoring HFRP rods. Wang et al. [11] evaluated the CFRP-steel bond performance depending on the toughness of adhesives and the boding thickness. Guo et al. [12] investigated the hygrothermal properties of pultruded FRP composites and evaluated the long-term hygrothermal evolution.

Polyurea is widely used as a waterproofing material because it has a fast curing time and is relatively insensitive to moisture. It is a material produced by a chemical reaction between an isocyanate-based polymer and an amine-based curing agent. It has high-tensile strength, high elongation, and excellent workability which is demonstrated by the ability to cure within 30 s at room temperature. Milled glass fiber with an average fiber length of 300 µm also has excellent tensile strength.

Discrete fiber-reinforced polyurea (DFRP) is a composite system used to simultaneously spray the polyurea and chopped glass fibers in a spray pattern. Its flexural and shear capacities, as well as ductility, have been investigated [13,14]. Glass fiber-reinforced polyuria (GFRPU) is a composite elastomer in which powdered glass fibers are mixed with liquid polyurea at a speed of 500 rpm or more to further improve the tensile strength of polyurea. The mechanical improvement of RC members sprayed with GFRPU has been investigated and the validity was evaluated [15,16]. Various studies on the application of the GFRPU reinforcement method for practical designs are in progress.

As a composite elastomer, GFRPU is easy to construct and economical compared to other reinforcing materials. It is a material with high-tensile strength and excellent ductility. This study was planned to verify the adequacy of reinforcement of RC columns using these properties. Thus, the purpose of this study was to investigate the reinforcement effect of GFRPU on the shear load-resisting capacity of nonseismically detailed RC columns. It evaluated the effectiveness of shear-strengthening techniques on various axial-load and shear-span ratios. Six RC column specimens were manufactured and tested under a cyclic uniaxial load and a constant axial load. The improvement of the lateral-reinforcement effect and energy-dissipation capacity was evaluated using experimental variables. The design method for the lateral-resistance ability of the GFRPU was presented by an empirical formula that reflected the experimental results.

## 2. Test Specimens and Loading Layout

### 2.1. Experimental Materials

The shear-load carrying capacity and deformation capacity of RC columns confined with sprayed GFRPU were investigated. The amount of glass fiber in a GFRPU mixture has a large influence on the tensile strength of the GFRPU and its constructability. In a previous study [11], it was reported that the optimal mixing ratio of glass fiber to polyurea required to obtain the maximum tensile strength was a weight ratio of 5%. Thus, we selected a weight ratio of 5% glass fiber for the GFRPU mixture.

The material properties of the milled glass fiber are summarized in Table 1. When the milled glass fiber was mixed with polyurea, the nozzles clogged due to separation or agglomeration. In order to overcome this, the rotation speed was adjusted to a speed of 500 rpm, which was established through repeated experiments, so as to apply the spray smoothly. The reactor was specifically manufactured to produce a GFRPU with uniform properties, by mixing the curing agent and liquid polyurea. The reactor was composed of a heating device made in Korea—to maintain the temperature above 65 °C—and a device for the continuous stirring of the glass fiber made in Korea. The tensile strength of the GFRPU was measured as 29 MPa, in accordance with KS F 4922.

The material used in the experiment included four ϕ100 × 200 mm concrete cylinders that were manufactured according to KS F 2403, and their compressive strength was measured during the test of the RC columns. The four-week average compressive strength of the concrete of the cylinder specimens was measured as 12 MPa for the column, and 27.8 MPa for the footing. However, the compressive strength of the concrete for the column was lower than the design strength despite the production order. It was initially designed to be 21 MPa, but the measured value showed lower compressive strength. In the subsequent experiments, the effect of the decreased concrete strength was taken into account. The failure mode and the influence on the axial-force ratio were investigated.

The effect of the axial load acting on the RC column specimens was determined to be greater than the load in the experiment plan because it had been planned using the concrete strength from the design. The initial axial-load ratios of 0.1 and 0.2 were raised to 1.75 and 2.75, respectively. Despite the lower-than-expected concrete strength, the strengthening effect of the GFRPU was still evaluated.

### 2.2. Specimens

The specimen was manufactured according to the standards in place before the seismic design standards were enacted, to verify the reinforcement effect of the GFRPU. The specimens were designed with insufficient reinforcement and nonseismic reinforcement details. Two control specimens were manufactured without any reinforcement material to evaluate the strengthening effect of the GFRPU. The specimens were sorted according to their shear-span ratio and axial-load ratio. The test variables were with and without the GFRPU reinforcement. The specimen was named by using H for a shear-span ratio of 5.5, S for 2.8, R when reinforced by GFRPU, U when unreinforced, and with axial-load ratios of 0.1 and 0.2. The H-series and S-series cantilevered specimens had shear spans of 1660 mm and 830 mm, respectively, and a constant cross-section of 300 × 300 mm. The specimens were reinforced longitudinally with four 16 mm deformed bars. The H- and S-series specimens were laterally reinforced with seven and five 10 mm deformed bars with a spacing of 300 mm, respectively. The coating thickness of the GFRPU was 5 mm. The axial-load ratio using the design concrete strength was designed to be 0.1 and 0.2. The lateral load was applied to the mid-depth of the upper slab. The GFRPU was sprayed onto the entire column. The details of the specimens are summarized in Figure 1 and Figure 2.

The details of the test setup are shown in Figure 3. A pseudodynamic test was performed to investigate the elastic–plastic hysteresis characteristics of the specimen. The pseudodynamic test method was adopted as a viable alternative to large-scale shaking table systems for evaluating structural performance. The loading was provided by the displacement control method that increased the drift level at a certain rate up to the displacement at failure. A hydraulic pressure device made in Korea with a capacity of 250 mm and a maximum load of 1000 kN was used. An axial load in a vertical direction was applied to the specimen by an actuator to maintain axial force.

The experiment was conducted by applying lateral load in laterally-positive and -negative directions. The behavior, lateral strength, ductility, and failure modes of the specimens were investigated according to the presence or absence of GFRPU reinforcement while loading. Strain gauges were attached to the longitudinal bars and tie bars of the specimens to measure the strain according to the load.

The lateral loading was planned using FEMA 461 and each step had two cycles in the loading history. Taking a loading history with $a_{n}=\Delta_{m}$ and $a_{1}=0.048\Delta_{m}$, the amplitude $a_{i+1}$ of the step i+1 is given by
(1)$$a_{i+1}=1.4a_{i}$$
where *a_i_* is the amplitude of the preceding step and *a_n_* is the amplitude of the step close to the target, Δ*_m_*. The loading history for the H- and S-series specimens is shown in Figure 4. 

## 3. Test Results and Discussion

### 3.1. Failure Mode

The initial crack of the unreinforced RC columns initiated at the lower end of the column. The high-axial-compressive load delayed the development of the cracks while the lateral load acted on the specimen. None of the specimens with nonseismic reinforcement details retained sufficient lateral-resisting capacity. In the case of the H-series specimen, as shown in Figure 5, the crack gradually moved upward, and the crack width increased with the increase in the lateral load. In the case of the S-series specimen, as shown in Figure 6, the flexural and diagonal tension cracks propagated on the upper side of the column. The crack width also gradually increased, and the specimens failed. It was identified that the column was not sufficiently fixed to the footing due to poor anchorage between both members. As a result, a large crack was observed along the joint.

In the case of the GFRPU-reinforced specimen, the appearance of cracks could not be visually confirmed due to the colored GFRPU coating. It was inferred that similar cracks occurred inside the coated surface; however, it was predicted that the propagation of the cracks was controlled and delayed by the axial load, the tensile strength, and the ductile capacity of the GFRPU. The GFRPU improved the anchorage capacity between the column and the footing due to the bond strength. It was determined that the GFRPU played an important role in controlling the failure of the column and enhancing its toughness.

The axial-load ratio was limited to 0.2 or less because the specimen was manufactured according to the RC column design criteria with nonseismic reinforcement details. However, the axial load acted according to the increased load ratio during the experiment because the specimens were produced by low compressive-strength concrete. The increased axial load also caused a delay in the development of cracks.

### 3.2. Cyclic Load and Deflection

The experimental results are summarized in Table 2. It was observed that the lateral-load-carrying capacity and ductile capacity were improved by the GFRPU reinforcement. The GFRPU reinforcement improved the shear strength as well as the ductility of the RC column. It was observed that the degree of GFRPU reinforcement was greatly affected by the shear-span ratio and the axial-load ratio. The decrease in the shear-span ratio provided the RC specimens with more distinct shear-reinforcement capacity and exhibited a clearer increase in the shear-resisting capacity. The shear strength also decreased with the increase in the axial-load ratio. It was predicted that the deterioration was caused by the P-Δ effect of the high-axial load and drift. In the H-series, the energy-dissipation capacity decreased as the axial-load ratio increased, but slightly increased in the case of the S-series. It was determined that this was because the column was stiffer due to the increase in the lateral-load-carrying capacity with the decrease in its height.

Figure 7 presents the cyclic load and deflection curves. As shown in the plots, the reinforcement effect of the GFRPU was analyzed. The transverse resistance capacity of the specimens was improved with a low shear-span ratio and high axial-load ratio. However, it was observed that the lateral-resistance capacity was slightly increased despite the reinforcement of the increase in the actual axial-load ratio. Crack propagation could not be observed after every cycle of lateral load applied. However, as the load increased or the load cycle increased, the crack propagated from the bottom to the top and the crack width became larger. The cracks were more distinct and wider in the S-series specimen than in the H-series specimen.

Considering the low compressive strength of the concrete, a large reinforcement effect from the GFRPU was observed in the specimens with a large shear-span ratio and an axial-load ratio of 0.1. The actual axial-load ratio of 0.2 corresponded to about 30.8% of the maximum axial load-carrying capacity of the RC column. The maximum shear load-carrying capacity of the S-series specimens was found to be greater than that of the H-series owing to the short shear span and large-lateral reinforcement. The H-series specimen with an axial-load ratio of 0.1 exhibited higher shear strength, of about 30%, than the specimen without reinforcement. The specimen with the axial-load ratio of 0.2 resulted in a higher load-carrying capacity, of about 4%, than the specimen with the axial-load ratio of 0.1. This was due to the increase in the lateral-resistance capacity. When the axial-load ratio was increased to 0.2, the degree of increase in the lateral load-resisting capacity was insignificant. Despite the GFRPU reinforcement, the shear-reinforcement capacity was slightly larger than the nonreinforced specimen. This phenomenon was likely to be due to the P-Δ effect between the displacement drift and the axial force.

Similar results were also observed for the S-series specimens. The shear load-carrying capacity of the specimens reinforced by the GFRPU with the axial-load force ratios of 0.1 and 0.2 increased by 18% and 4%, respectively. If the axial-load ratio was greatly increased, the lateral-resistance capacity of the reinforcement material was lowered, and ultimately the reinforcement effect was found to be insignificant. From the load and deflection relationship, it was established that the lateral-reinforcement effect and the degree of increase from the GFRPU could be evaluated by the axial-load ratio and the shear-span ratio.

Figure 8 presents the relationship between the lateral force and drift ratio. It was observed that the peak load was reached at a similar drift ratio. In these figures, the H-series specimens had greater deformation capacity than the S-series, but the transverse resistance capacity was rather low. In the case of each series, it was observed that the deformation capacity as well as the lateral-resistance capacity were improved by the GFRPU reinforcement. The lateral-resistance capacity of the specimen with nonseismic reinforcement details was greatly improved by the GFRPU reinforcement for the same axial-load ratio. In the case of the reinforced specimen with an axial-load ratio of 0.1, the transverse-resistance capacity was the largest, but the loading capacity decreased rapidly after the maximum load. It was considered that the column was slightly damaged by the high-axial load. In this experiment, a higher axial load than was expected acted on the RC column. Thus, if the axial-load ratio was increased more than 0.2, the transverse resistance and deformation capacity were slightly improved despite the GFRPU reinforcement.

### 3.3. Energy Dissipation

Energy dissipation is related to permanent deformation and damage within a concrete member as a result of an irreversible dissipation process. Dissipated energy was measured by estimating the hysteretic curve area surrounded by the second cycle in the cyclic lateral load and displacement-drift curves. As shown in Table 2.

The load-carrying capacity of the S-series specimen was larger than that of the H-series specimen, and the deformation capacity was inverse. The deformation capacity of the H-series was greatly improved by the GFRPU reinforcement. The energy-dissipation capacity of the H-series specimens decreased with the increase in the axial-load ratio. It was determined that the decrease was a result of the P-Δ effect. It was observed that the energy-dissipation capacity of the S-series specimens slightly increased because the specimens were stiffer than the H-series specimens, but the deformation capacity did not increase significantly. It was determined to be improved because of the restraint and lateral-deformation capacity of the GFRPU against the damage of concrete. Despite the increase in the axial-load ratio, no significant change in the energy-dissipation capacity was shown. It was observed that the dissipated energy capacity was greatly improved by the GFRPU reinforcement of the H-series specimen with the large column height. However, it was predicted that the dissipated energy capacity was not significantly improved by the P-Δ effect with the increase in the axial-load ratio.

### 3.4. Empirical Shear Strength

In addition to the previous study [11], this experiment was conducted with the test variables of shear-span ratio and axial-load ratio. An empirical formula for estimating the improvement of shear strength by the GFRPU including the test parameters was developed. It was known from the results of this study that the effect of the axial-load ratio was relatively smaller than the shear-span ratio. Therefore, the empirical formula modified the effect of the axial-load ratio and the shear-span ratio.

The equation for predicting the shear strength carried by the GFRPU [11] can be expressed by
(2)$$V_{f}=\upsilon \lambda_{f}f_{t}bd$$
where $\upsilon =\frac{1.5\left(0.403-1.053\frac{P}{A_{g}f_{c}^{\prime }}+0.176\frac{a}{h}\right)}{\sqrt{\lambda_{f}}+1.207}$ and $\lambda_{f}=\frac{2t_{f}b_{f}}{bh}\frac{f_{f}}{f_{t}}$. a and d are the shear span and effective depth of the section, respectively; P is the axial load; $A_{g}$ is the gross area of the section; b denotes the width of the section; h is the column height; $t_{f}$ and $b_{f}$ denote the thickness and width of the sprayed GFRPU, respectively; $f_{f}$ is the tensile strength of GFRPU; and $f_{t}$ is the flexural tensile strength of the concrete.

It was found that the shear-span ratio had a greater effect on the shear strength than the axial-force ratio. The coefficient υ was modified to reflect the relative relationship using the experimental results as follows:(3)$$\upsilon =\frac{1.5\left(0.403-3.5\frac{P}{A_{g}f_{c}^{\prime }}+0.14\frac{a}{h}\right)}{\sqrt{\lambda_{f}}+1.207},\;\lambda_{f}=\frac{2t_{f}b_{f}}{bh}\frac{f_{f}}{f_{t}}.$$

The coefficient of the axial-load ratio of 1.053 was changed to 3.5, and the coefficient of the shear-span ratio of 0.176 was reduced to 0.14.

Table 2 compares the maximum shear load obtained by the experiment with the shear strength estimated by Equation (3). As shown in this table, it can be observed that the load-carrying capacity of the GFRPU in the empirical formula is within the range of 98.5% to 102.1% of the experimental results. This indicates that the GFRPU design can be performed by the proposed formula.

## 4. Conclusions

In this study, the shear reinforcement effect of the sprayed GFRPU on RC columns was investigated using a cyclic load test. The shear-span ratio and axial-load ratio were used as experimental variables. The specimens were manufactured according to the design standards of nonseismic reinforcement.

The GFRPU reinforcement greatly improved the shear strength and energy-dissipation capacity of the RC columns. It was observed that the shear strength was improved within the range of 4–30% depending on the shear-span ratio and the axial load. A higher axial load than expected acted on the column because the concrete was manufactured to a lower strength than the strength specified in the design. It was determined that the shear-strength and energy-dissipation capacity would insignificantly increase if the axial-load ratio was greater than 0.2. This was due to the potential damage by the action of the high-axial load. An empirical formula to describe the degree of improvement of the shear strength caused by the GFRPU was proposed including the shear-span ratio and the axial-force ratio. It was observed that the proposed formula could properly estimate the degree of reinforcement required to apply this to a practical strengthening design. It is necessary to investigate the physical properties of GFRPU and to supplement the empirical formula through more experiments on the parameters affecting the shear strength of RC columns.

## Figures and Tables

**Figure 1 polymers-14-03596-f001:**
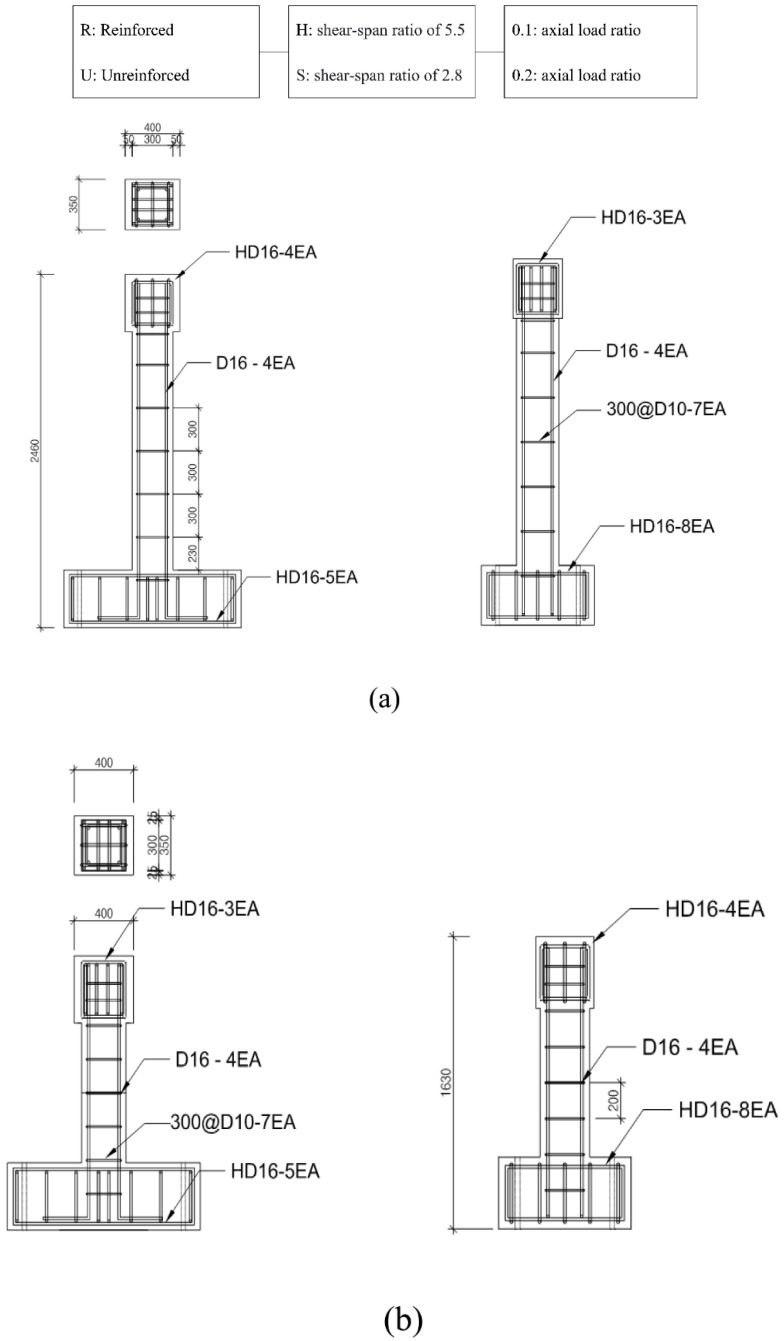
Reinforcement placement of (**a**) H-series and (**b**) S-series (unit: mm). 300@ means 300 mm spacing. 300@ means 300 mm spacing.

**Figure 2 polymers-14-03596-f002:**
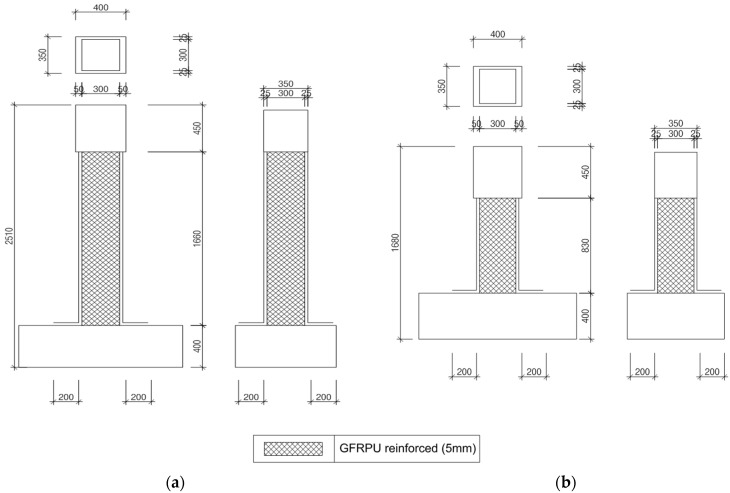
Reinforcement of GFRPU of (**a**) H-series and (**b**) S-series (unit: mm).

**Figure 3 polymers-14-03596-f003:**
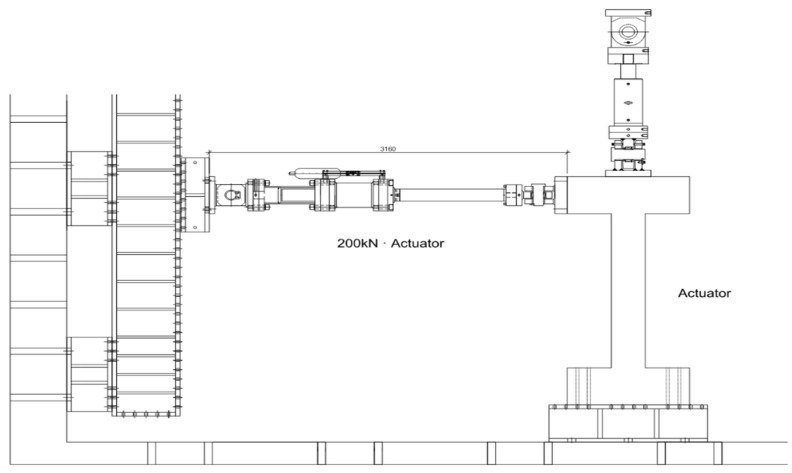
Experimental equipment layout.

**Figure 4 polymers-14-03596-f004:**
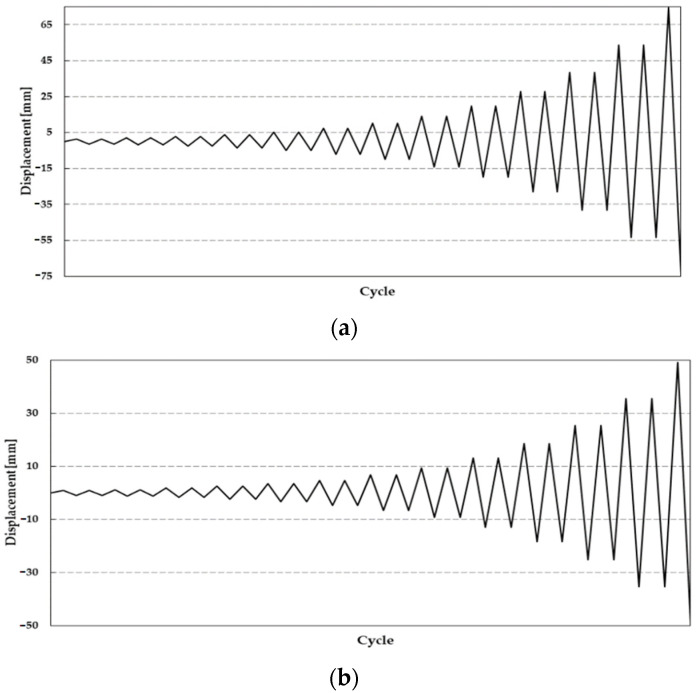
Cyclic loading pattern of (**a**) H-series and (**b**) S-series.

**Figure 5 polymers-14-03596-f005:**
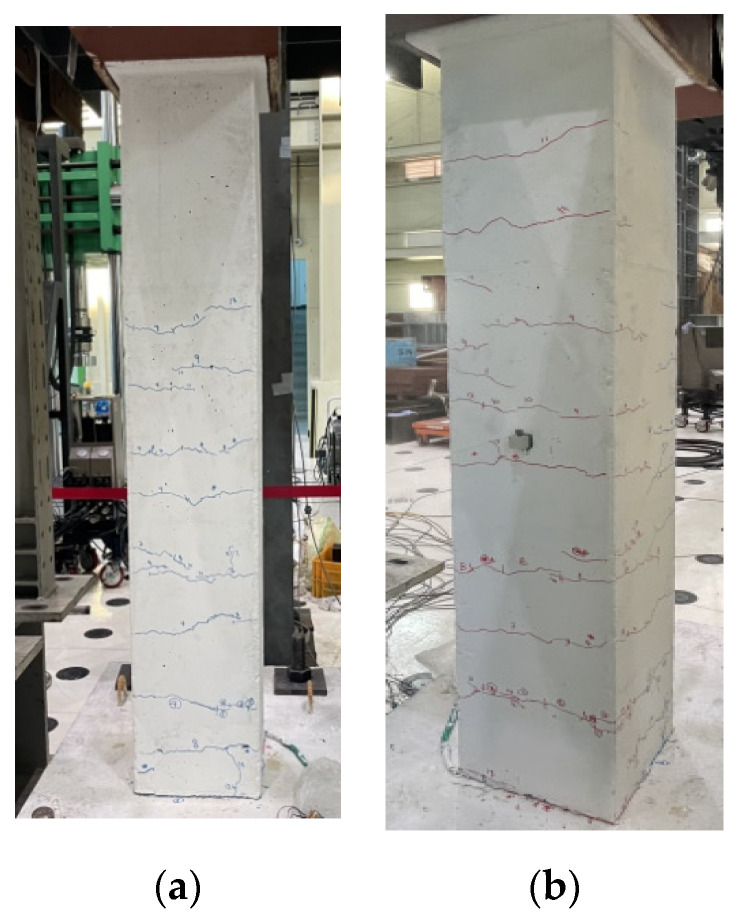
The failure mode of unreinforced H-series specimen. (**a**) front view, (**b**) side view.

**Figure 6 polymers-14-03596-f006:**
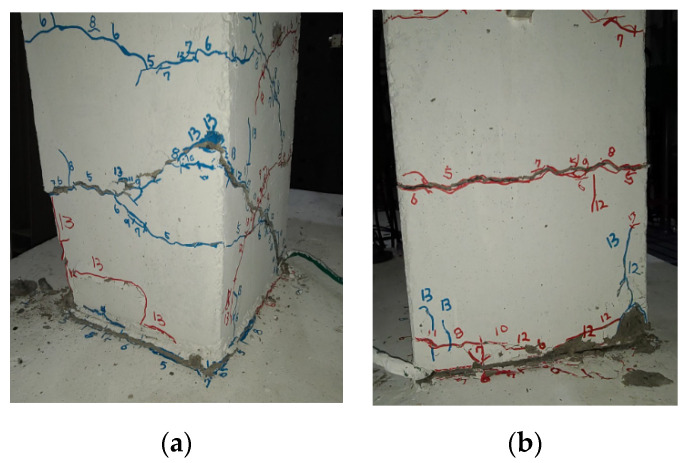
The failure mode of unreinforced S-series specimen. (**a**) front view, (**b**) side view.

**Figure 7 polymers-14-03596-f007:**
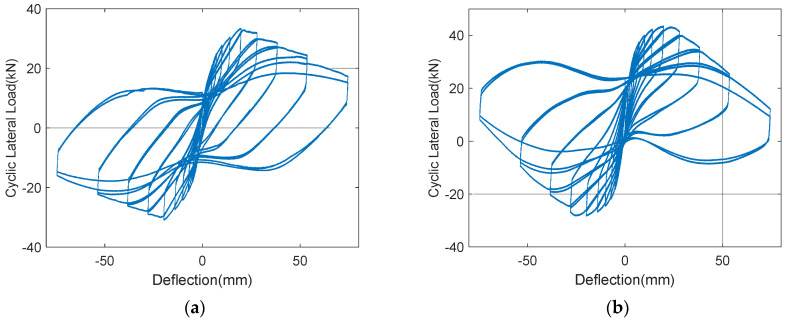
Load-displacement curves of (**a**) U-H-0.1, (**b**) R-H-0.1, (**c**) R-H-0.2, (**d**) U-S-0.1, (**e**) R-S-0.1, (**f**) R-S-0.2.

**Figure 8 polymers-14-03596-f008:**
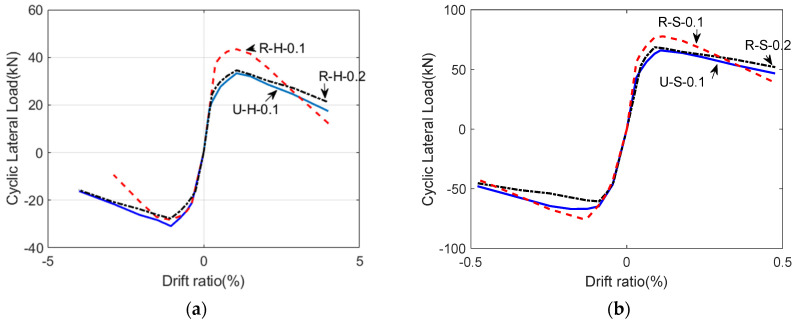
Lateral load-drift ratio curves of (**a**) H-Series and (**b**) S-series.

**Table 1 polymers-14-03596-t001:** The physical properties of milled glass fiber.

	Unit	Value
average filament diameter	µm	13.5
average filament length	µm	300
density	g/cm^3^	0.58
moisture content	%	Below 0.05

**Table 2 polymers-14-03596-t002:** Summary of experimental results.

Specimen	*V*_exp_*max*_ (kN)		*V*_anal_*max*_ (kN)	Energy Dissipation (kN·m)
	Pos.	Neg.		
U-H-0.1	33.4	−31.0		1.05
R-H-0.1	43.6	−28.4	44.5 (102.1%)	2.39
R-H-0.2	34.6	−27.7	34.3 (99.1%)	1.085
U-S-0.1	66.3	−67.0		1.8
R-S-0.1	78.1	−75.2	76.9 (98.5%)	1.67
R-S-0.2	68.9	−61.1	69.9 (101.5%)	1.91

*V*_exp_*max*_ and *V*_anal_*max*_ represent the experimental and analytical maximum shear-resisting capacity, respectively.

## Data Availability

The data used to support the findings of this study are included within the article.

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
