# Peer review of "Shear Strengthening by GFRPU on an RC Column with Nonseismic Reinforcement Details"

_polymers, 2022, doi:10.3390/polym14173596_

Round 1
Reviewer 1 Report
The reviewer consider the study interesting and consider that can be published in this journal after a few improvements.
- regarding the load-displacement curves presented in Figure 7 identify the level of damage that occurred after each cycle.
- The Figure 8 show the lateral load-drift ratio curves of H and S - Series and can be observed that the level of load reached is higher for S-Series. This is due to the length of the column that is lower than the H-Series.
The authors need to add a normalized loading taking into account the column length. In order to see clearly the influence of column length on the maximum load.
Author Response
Reviewer 1
Thank you for your invaluable comment. We considered your comment and reply as follows:
The reviewer consider the study interesting and consider that can be published in this journal after a few improvements.
1-1. regarding the load-displacement curves presented in Figure 7 identify the level of damage that occurred after each cycle.
-à It was added by Reviewer 1-1. Cracks propagation could not be observed after every cycle of lateral load applied. However, as the load increased or the load cycle increased, the crack propagated from the bottom to the top, and the crack width became larger. The cracks were more distinct and wider in the S series specimen than in the H series specimen.
1-2. The Figure 8 show the lateral load-drift ratio curves of H and S - Series and can be observed that the level of load reached is higher for S-Series. This is due to the length of the column that is lower than the H-Series.
1-3.The authors need to add a normalized loading taking into account the column length. In order to see clearly the influence of column length on the maximum load.
-à We agree with you. The different behavior between H and S series specimens is affected by the column length. In order to explain the effect of the column length, the shear-span ratio normalized by the depth of the column section was utilized.
Reviewer 2 Report
The subject of manuscript is interesting, however structure of manuscript is not well-constructed. All presented figures have very good quality and they prove their point. Results are sufficient and this manuscript may be contribute to science.
The paper is written in good English. The manuscript is easily readable concerning language, style and presentation.
According to my scientific evaluation, this manuscript can be published after minor revision.
Detailed comments:
The references are appropriate and up to date. However, due to the prevalence of research on the shear strength and energy dissipation capacity of the reinforced concrete columns, the scope of the reviews seems very limited. The authors only found 13 references.
According to the "Introduction" section the authors have investigated the reinforcement effect of GFRPU on the shear load-resisting capacity of non-seismically detailed RC columns. My main question is, what is the actual scientific novelty of the article.
The authors presented physical property of milled glass fibre. However, the details of the composition of the concrete cylinders was not described.
It is desirable to clarify the procedure of "a pseudo dynamic test" for non-specialist readers.
The structure of this manuscript needs to be revised. It is suggested to present experimental procedures and results in separate sections.
There are many formatting problems in the manuscript.
Results are well presented in this manuscript, however, discussions are not sufficient in many places. Please discuss the results with more current references, which compare the results obtained by the authors with other studies carried out by other researchers.
The conclusions section is very general. It should not be a summary of your study or an extension of the discussion. The conclusions section should illustrate the mechanistic links of findings obtained under applied treatments. The authors should avoid repeating what has already been presented in results and discussion. It is also suggested to add quantitative conclusions.
In addition, it is essential to indicate future lines of research.
Author Response
Reviewer 2.
Thank you for your invaluable comment. We considered your comment and reply as follows:
The subject of manuscript is interesting, however structure of manuscript is not well-constructed. All presented figures have very good quality and they prove their point. Results are sufficient and this manuscript may be contribute to science.
The paper is written in good English. The manuscript is easily readable concerning language, style and presentation.
According to my scientific evaluation, this manuscript can be published after minor revision.
Detailed comments:
2-1. The references are appropriate and up to date. However, due to the prevalence of research on the shear strength and energy dissipation capacity of the reinforced concrete columns, the scope of the reviews seems very limited. The authors only found 13 references.
-à Four more references were added including the references recommended by another reviewer.
2-2. According to the "Introduction" section the authors have investigated the reinforcement effect of GFRPU on the shear load-resisting capacity of non-seismically detailed RC columns. My main question is, what is the actual scientific novelty of the article.
-à It seems to be a good question. As a composite elastomer, GFRPU is easy to construct and economical compared to other reinforcing materials. It is a material with high tensile strength and excellent ductility. This study was planned to verify the adequacy of reinforcement of RC columns using these properties.
2-3. The authors presented physical property of milled glass fibre. However, the details of the composition of the concrete cylinders was not described.
-à GFRPU is a reinforcing material used to improve strength by applying it to the concrete surface rather than mixing it in the concrete. Therefore, only the components of GFRPU were introduced.
2-4. It is desirable to clarify the procedure of "a pseudo dynamic test" for non-specialist readers.
-à The pseudo dynamic test method was adopted as viable alternative to large scale shaking table systems for evaluating the structural performance.
2-5. The structure of this manuscript needs to be revised. It is suggested to present experimental procedures and results in separate sections.
2-6. There are many formatting problems in the manuscript.
Results are well presented in this manuscript, however, discussions are not sufficient in many places. Please discuss the results with more current references, which compare the results obtained by the authors with other studies carried out by other researchers.
The conclusions section is very general. It should not be a summary of your study or an extension of the discussion. The conclusions section should illustrate the mechanistic links of findings obtained under applied treatments. The authors should avoid repeating what has already been presented in results and discussion. It is also suggested to add quantitative conclusions.
In addition, it is essential to indicate future lines of research.
-à It seems to be good comment. We carefully considered. The subject regarding the reinforcement by the GFRPU was hardly found any research results. It is incomparable to the experiment with other reinforcing materials. Section 2 has been written by dividing it into 2 and 3. Quantitative assessment and future work were added to the conclusion.

Reviewer 3 Report
The authors have investigated the shear strengthening by GFRPU on a RC column with non-seismic reinforcement. The following comments can further improve the quality of the paper.
1. The current abstract does not convey some important information and key results about this paper. Please provide some quantitative analysis results.
2. In the second part of the introduction, the authors introduce three strengthening methods. Furthermore, the FRP strengthening method can significantly improve the shear performance, toughness and energy dissipation of concrete columns. However, there is a lack of the information on the types, performance, advantages and engineering applications of FRP. This information can help readers better understand FRP and its engineering applications. Please review some latest research on FRP. Construction and Building Materials, 2022, 315: 125710. Thin-walled structures, 2021, 158: 107176. Materials and Structures, 2020, 53: 73.
3. The information of GFRPU is not enough. It is suggested to provide the advantages, performance and relevant engineering applications.
4. In the last part of the introduction, it is suggested that the authors can further put forward the contribution and innovation of this work by describing the current research work.
5. In Section 2.2, please provide some basis to choose the specimens with the different shear-span ratio and axial-load ratios.
6. Please improve the clarity of Fig. 2 and Fig. 3. Some details should be further enlarged to increase its clarity.
7. The second part should focus on the materials and test methods related to the experiment, and the part about the results and discussion (from the part of 2.3) should be placed in the third part. It is suggested that the authors rewrite the experiment and results according to the above comments.
8. Fig. 5 and Fig. 6 show the failure modes of unreinforced specimens, and the failure modes of the samples of reinforced specimen should also be given for comparison and analysis.
9. Please further check and standardize the drawing of the tables and figures.
10. The conclusions should be further rewritten, only including some key information.
Author Response
Reviewer 3
Thank you for your invaluable comment. We considered your comment and reply as follows:
The authors have investigated the shear strengthening by GFRPU on a RC column with non-seismic reinforcement. The following comments can further improve the quality of the paper.
3-1. The current abstract does not convey some important information and key results about this paper. Please provide some quantitative analysis results.
-à We added quantitative results. The H-series and S-series specimens reinforced by the GFRPU with an axial-load ratio of 0.1 exhibited higher shear strength, about 30% and 18%, than the specimens without reinforcement. The shear load-carrying capacity of the S-series specimen with the axial-load force ratio of 0.2 increased by 4%.
3-2. In the second part of the introduction, the authors introduce three strengthening methods. Furthermore, the FRP strengthening method can significantly improve the shear performance, toughness and energy dissipation of concrete columns. However, there is a lack of the information on the types, performance, advantages and engineering applications of FRP. This information can help readers better understand FRP and its engineering applications. Please review some latest research on FRP. Construction and Building Materials, 2022, 315: 125710. Thin-walled structures, 2021, 158: 107176. Materials and Structures, 2020, 53: 73.
-à We added the properties of the FRP and some references including the recommended references.
. FRP is a composite material made of a polymer matrix reinforced with fibers such as glass, carbon, or aramid.
Shaoqing and Kono [9] investigated the shear contribution of the FRP sheets to prevent the brittle fracture of the column. Li et al. [10] developed a hybrid fiber reinforced polymer (HFRP) to replace the steel cables and investigated the failure modes of anchoring HFRP rods. Wang et al. [11] evaluated the CFRP-steel bond performance depending on the toughness of adhesives and the boding thickness. Guo et al. [12] investigated the hygrothermal properties of pultruded FRP composites and evaluated the long-term hygrothermal evolution.
3-3. The information of GFRPU is not enough. It is suggested to provide the advantages, performance and relevant engineering applications.
-à It was added.
As a composite elastomer, GFRPU is easy to construct and economical compared to other reinforcing materials. It is a material with high tensile strength and excellent ductility. This study was planned to verify the adequacy of reinforcement of RC columns using these properties.
3-4. In the last part of the introduction, it is suggested that the authors can further put forward the contribution and innovation of this work by describing the current research work.
-à It was included in the comment of 3-3.
3-5. In Section 2.2, please provide some basis to choose the specimens with the different shear-span ratio and axial-load ratios.
-à Since the shear strength of the RC member is greatly affected by the shear-span ratio, it was designed as a member with a large effect of shear and flexure. The axial load ratio was set according to the seismic performance evaluation guidelines for existing facilities published by the Ministry of Land, Infrastructure and Transport. The guideline established the axial load ratio of 0.1 and 0.6, but in this study, 0.1and 0.2 were selected and the values between them were interpolated linearly.
3-6. Please improve the clarity of Fig. 2 and Fig. 3. Some details should be further enlarged to increase its clarity.
-à They were enlarged.
3-7. The second part should focus on the materials and test methods related to the experiment, and the part about the results and discussion (from the part of 2.3) should be placed in the third part. It is suggested that the authors rewrite the experiment and results according to the above comments.
-à It was written separately in Sections 2 and 3 as you mentioned.
3-8. Fig. 5 and Fig. 6 show the failure modes of unreinforced specimens, and the failure modes of the samples of reinforced specimen should also be given for comparison and analysis.
-à It’s impossible to observe the occurrence and development of the cracks because the GFRPU was sprayed on the concrete surface and was coated by green color.
In the case of the GFRPU-reinforced specimen, the appearance of cracks could not be visually confirmed due to the colored GFRPU coating. It was inferred that similar cracks occurred inside the coated surface; however, it was predicted that the propagation of the cracks was controlled and delayed by the axial load, the tensile strength and the ductile capacity of the GFRPU.
3-9. Please further check and standardize the drawing of the tables and figures.
-à It was checked and modified. If necessary, we will redraw them to meet the guidelines.
3-10. The conclusions should be further rewritten, only including some key information.
-à It was modified.

Round 2
Reviewer 3 Report
It is recommended to accept this paper.